# Blood Pressure and Heart Rate Responses following Dietary Protein Intake in Older Men

**DOI:** 10.3390/nu14091913

**Published:** 2022-05-03

**Authors:** Avneet Oberoi, Caroline Giezenaar, Kylie Lange, Karen L. Jones, Michael Horowitz, Ian Chapman, Stijn Soenen

**Affiliations:** 1Adelaide Medical School, Centre of Research Excellence in Translating Nutritional Science to Good Health, The University of Adelaide, Royal Adelaide Hospital, Adelaide 5000, Australia; avneet.oberoi@adelaide.edu.au (A.O.); kylie.lange@adelaide.edu.au (K.L.); karen.jones@adelaide.edu.au (K.L.J.); michael.horowitz@adelaide.edu.au (M.H.); ian.chapman@adelaide.edu.au (I.C.); 2Riddett Institute, Massey University, Palmerston North 9430, New Zealand; c.giezenaar@massey.ac.nz; 3Faculty of Health Sciences and Medicine, Bond University, Gold Coast 4229, Australia

**Keywords:** whey protein, blood pressure diet, heart rate, aging

## Abstract

Postprandial hypotension (PPH) occurs frequently in older people >65 years old. Protein-rich supplements, particularly whey protein (WP), are increasingly used by older people for various health benefits. We have reported that 70 g WP drinks cause significant, and in some cases marked, falls in blood pressure (BP) in older men. The effects of lower, more widely used, doses (~30 g) on systolic (SBP) and diastolic (DBP) blood pressure and heart rate (HR) are not known. In a randomized order, eight older men (age: 72 ± 1 years; body mass index (BMI): 25 ± 1 kg/m^2^) after overnight fast ingested a drink containing (i) a non-caloric control (~2 kcal), (ii) 30 g of whey protein (120 kcal; ‘WP30’), or (iii) 70 g of whey protein (280 kcal; ‘WP70’). The BP and HR were measured in this pilot study with an automated device before and at 3-min intervals for 180 min following drink ingestion. Drink condition effects were determined by repeated-measures ANOVA. The SBP decreased after both WP drinks compared to the control (*p* = 0.016), particularly between 120 and 180 min, with no difference in the effects of WP30 and WP70. The SBP decreased by ≥20 mmHg in more than 50% of people after both WP drinks (WP30: 63%; WP70: 75%) compared to 38% after the control. The maximum fall in the SBP occurred during the third hour, with the nadir occurring latest after WP70. The DBP decreased non-significantly by several mmHg more after the WP drinks than after the control. The maximum HR increases occurred during the third hour, with the greatest increase after WP70. The SBP decreased after both WP drinks compared to the control, with the effects most evident between 120 and 180 min. Accordingly, ingestion of even relatively modest protein loads in older men has the potential to cause PPH.

## 1. Introduction

Human aging is associated with reductions in skeletal muscle mass and strength, which are associated with increased rates of falls and nursing home admissions, as well as other adverse outcomes [1,2,3,4,5,6,7,8]. Falls may also be caused by dizziness and/or syncope due to postprandial hypotension (PPH), a substantial reduction in blood pressure (BP) caused by ingestion of nutrients [9,10], which has been defined as a decrease in systolic blood pressure (SBP) of ≥20 mmHg for ≥30 min within 120 min of consumption of a meal [11]. PPH occurs frequently in older people and is associated with increased morbidity and mortality [11].

A strategy increasingly adopted to prevent or treat under-nutrition, weight loss, and sarcopenia in older people is to increase consumption of high-energy, protein-rich supplements [12]. These supplements are often rich in whey protein [13], a component of milk, which is high in essential amino acids, particularly leucine, which are rapidly digested to increase postprandial amino acid availability and stimulate muscle protein accretion [14,15].

Oral ingestion of whey protein or whey-protein-rich supplements has the potential to reduce the BP to a degree that is harmful in some older people, predisposing to falls and other adverse effects. Compared to younger adults, older people exhibit greater decreases in the BP after meals [16]. Ingestion of all three macronutrients (carbohydrate, protein, fat) decreases the post-prandial BP in older people [17,18]. Carbohydrate and protein lower the BP to a similar degree, although the fall in the BP may occur earlier after carbohydrate ingestion than protein ingestion [17]. The hypotensive effects of protein are likely to be mediated by amino acids produced by digestion, explaining the latency and time of onset of changes in the BP and HR after protein loads. We have recently reported that ingestion of a 70 g whey protein drink is associated with a substantial decrease in the BP in healthy older men; the majority of older men studied had a decrease in the systolic BP (SBP) of 20 mmHg or more, with the greatest reduction occurring 2–3 h after drink ingestion [18].

Overall, 70 g of protein is more than most older people ingest at one time; amounts of ~30 g are probably sufficient to induce muscle synthesis in older people, particularly if ingested twice a day [19]. It is not clear whether the hypotensive effects of whey protein drinks are dose dependent in older people and whether whey doses less than 70 g also cause a substantial fall in the BP. We have reported that ingestion of a whey protein drink as a preload in combination with guar in a much lower dose of 16.4 g has no effects on the BP in healthy older people for up to 2 h [20]. Two hours may not, however, have been long enough to detect the maximum hypotensive effects of the drink in that study.

This study compared the effects over 3 hours of 30 g and 70 g whey protein drinks on the BP and heart rate (HR) in older men. This pilot study is a subset (*n* = 8 males) of a larger study and represents an analysis of secondary outcomes measured in men (BP and HR) in a previously published study [21] that described the effect of orally ingested whey protein on energy intake, gastric emptying, and plasma gut-hormone concentrations in older men (*n* = 8) and women (*n* = 8).

## 2. Materials and Methods

### 2.1. Subjects

Eight older (mean age: 73 ± 1 years; body weight: 77 ± 4 kg; body mass index (BMI): 26 ± 1 kg/m^2^) men were recruited by advertisement.

Exclusion criteria included alcohol intake of >2 standard drinks on >5 days per week; smoking; intake of any illicit substance; use of prescribed or non-prescribed medications that may affect appetite, body weight, gastrointestinal function, or energy metabolism; being vegetarian; known lactose intolerance or food allergies; epilepsy; gallbladder, pancreatic, cardiovascular, or respiratory diseases; significant gastrointestinal symptoms (abdominal pain, gastro-esophageal reflux, diarrhea, or constipation) or surgery; any other illness deemed significant by the investigator; low plasma ferritin levels; donation of blood in the 12 weeks prior to the study days; undernourished condition (score < 24 on the Mini Nutritional Assessment [22]); or depression (score ≥ 11 on the Geriatric Depression Questionnaire [23]), impaired cognitive function (score < 25 on Mini Mental State [24]), and inability to comprehend the study.

The Royal Adelaide Hospital Human Research Ethics Committee approved the protocol, which was conducted in accordance with the Declaration of Helsinki. The study was registered with the Australian New Zealand Clinical Trial Registry (www.anzctr.org.au (accessed on 15 October 2021; registration number ACTRN12612000941864). All subjects provided written informed consent prior to their study inclusion.

### 2.2. Protocol

The protocol was similar to that of our previous study comparing older and younger men [13]. Each participant was studied on 3 occasions, separated by 3–14 days, to determine the effects of drinks (~450 mL) containing (i) a non-caloric control (~2 kcal; ‘C’), (ii) 30 g of whey protein (120 kcal; ‘WP_30_′), or (iii) 70 g of whey protein (280 kcal; ‘WP_70_’) on the SBP, DBP, and HR in a randomized (using the method of randomly permuted blocks; www.randomization.com; accessed on 3 August 2018), double-blind, cross-over design.

The drinks were prepared by dissolving whey protein isolate (Fonterra Co-Operative Group Ltd., Palmerston North, New Zealand) in varying volumes of demineralized water and diet lime cordial (Bickford’s Australia Pty Ltd., Salisbury South, SA, Australia), to achieve the desired composition, by a research officer who was not involved in the data analysis on the morning of the study day. The drinks were matched for taste and served in a covered cup so that both the subject and the investigator were blinded to the treatment.

Subjects consumed a standardized meal (beef lasagne (McCain Foods Pty Ltd., Wendouree, Victoria, Australia), ~591 kcal) on the night before each study day at ~1900 h. They were instructed to fast overnight ~12 h, taking no solids and liquids except water.

### 2.3. Measurements

Subjects were seated in an upright position on a wooden chair with arms, where they remained throughout the study day. They were asked to sit quietly for at least 15 min, and the BP was measured 3 times at 3-min intervals using an automatic BP-measuring device (DINAMAP ProCare 100; GE Medical Systems, Milwaukee, WI, USA). Subjects were then instructed to consume the drink within 2 min, and the BP and HR were measured every 3 min for a further 180 min. When SBP measurements between consecutive samples varied by ≥10 mmHg, repeat measurements were taken.

### 2.4. Data and Statistical Analysis

Responses to the drinks were calculated for the first, second, and third hour following drink ingestion as the mean of individual measurements (mean 0–60 min, mean 60–120 min, mean 120–180 min, respectively). All data are presented as mean values ± standard error of the mean (SEM), and statistical significance was accepted at *p* < 0.05. Statistical analyses were performed using SPSS software (version 24; IBM, Armonk, NY, USA). The maximum decrease from baseline was calculated as the difference between the minimum value and baseline and the time of nadir that was analyzed for the SBP, DBP, and HR.

The baseline blood pressure was calculated as an average of −9-, −6-, and −3-min readings. T = 0 min refers to the point immediately after drink consumption. Mixed effects model analysis including treatment as a fixed effect, and an unstructured covariance structure was used to determine the effect of protein load on the SBP, DBP, and HR baseline levels and each of the three mean outcomes, along with the maximum decrease and the time to nadir. When significant treatment effects were present, Bonferroni-corrected post hoc tests were performed to determine which specific drink conditions differed.

## 3. Results

The study protocol was well tolerated by all subjects.

### 3.1. Systolic Blood Pressure (SBP)

The baseline SBP was comparable on the 3 study days (C: 133 ± 4 mmHg; WP30: 132 ± 4 mmHg; WP70: 134 ± 5 mmHg; *p* = 0.95). During the first 2 hours, SBPs after all drinks were not significantly different (mean 0–60 min, mean 60–120 min; C: 132 ± 6, 129 ± 6 mmHg, respectively; WP30: 127 ± 4, 124 ± 5 mmHg, respectively; WP70: 134 ± 5, 131 ± 6 mmHg, respectively; *p* = 0.12, *p* = 0.09, respectively). In contrast, between 120 and 180 min, the SBP was lower after both whey protein drinks than after the control drink (*p* = 0.016; C vs. WP30 *p* = 0.02; C vs. WP70 *p* = 0.20), with mean SBP changes from baseline of −10 mmHg after both whey protein drinks during the third hour compared to 0 mmHg after the control drink (mean 120–180 min; C: 133 ± 7 mmHg; WP30: 122 ± 6 mmHg; WP70: 125 ± 5 mmHg; Figure 1A, Table 1).

Maximum decreases in the SBP from baseline (C: −15 ± 2 mmHg; WP30: −24 ± 2 mmHg; WP70: −25 ± 4 mmHg; *p* = 0.02) occurred from ~2 h after drink ingestion onward, with the nadir (C: 118 ± 5 mmHg; WP30: 108 ± 5 mmHg; WP70: 110 ± 4 mmHg) occurring latest after the 70 g whey protein drink (C: 90 ± 8 min; WP30: 111 ± 22 min; WP70: 136 ± 11 min; *p* = 0.08). The SBP decreased by ≥20 mmHg in more than 50% of participants after both whey protein drinks (C: 3/8 (38%); WP30: 6/8 (75%); WP70: 5/8 (63%)).

### 3.2. Diastolic Blood Pressure (DBP)

The baseline DBP was not different on the study days (C: 80 ± 2 mmHg; WP30: 76 ± 2 mmHg; WP70: 80 ± 3 mmHg; *p* = 0.35). The mean fall in the DBP was greater after 30 g and 70 g whey protein drinks than after the control drink, which did not achieve statistical significance during any period. Mean decreases in the DBP from baseline during the third hour were −6/−8 mmHg after WP30/WP70 compared to −4 mmHg after the control drink (mean 0–60 min, mean 60–120 min, mean 120–180 min; C: 77 ± 7, 75 ± 3, 75 ± 3 mmHg, respectively; WP30: 73 ± 2, 71 ± 3, 70 ± 3 mmHg, respectively; WP70: 75 ± 3, 72 ± 3, 72 ± 3 mmHg, respectively; *p* = 0.23, *p* = 0.41, *p* = 0.20, respectively; Figure 1B, Table 1). Maximal changes in the DBP from baseline (C: −13 ± 2 mmHg; WP30: −15 ± 1 mmHg; WP70: −17 ± 2 mmHg; *p* = 0.13) occurred ~1.5–2 h after drink ingestion, with the nadir occurring latest after the 70 g whey protein drink (C: 89 ± 19 min; WP30: 98 ± 19 min; WP70: 108 ± 11 min; *p* = 0.74).

### 3.3. Heart Rate (HR)

The baseline HR was not different between the study days (C: 61 ± 3 bpm; WP30: 58 ± 2 bpm; WP70: 57 ± 2 bpm; *p* = 0.56). The HR was higher after the whey protein drinks when compared to the control drink during the last 2 hours (mean 0–60 min, mean 60–120 min, mean 120–180 min; C: 55 ± 3, 55 ± 2, 55 ± 2 bpm, respectively; WP30: 58 ± 3, 60 ± 3, 58 ± 3 bpm, respectively; WP70: 59 ± 3, 60 ± 3, 62 ± 3 bpm, respectively; *p* = 0.05, *p* = 0.007, *p* = 0.005, respectively), with the greatest increase after WP70 and during the third hour (change from baseline to 120–180 min: C: −5 ± 1 bpm; WP30: 0 ± 1 bpm; WP70: +5 ± 1 bpm; *p* < 0.05; Figure 1C, Table 1). The maximal increase in the HR from baseline (C: 2 ± 1 bpm; WP30: 6 ± 1 bpm; WP70: 10 ± 2 bpm; *p* = 0.002) occurred ~1.5–2 h after drink ingestion (C: 71 ± 25 min; WP30: 112 ± 17 min; WP70: 120 ± 17 min; *p* = 0.181).

## 4. Discussion

We demonstrated that healthy older men exhibit a decrease in the SBP after ingestion of both 30 g and 70 g whey protein drinks, which was similar in degree after both whey protein drinks and the greatest between 120 and 180 min after their ingestion. Previously, we reported that protein, when administered directly into the duodenum and thereby bypassing gastric effects, lowers blood pressure comparably to glucose and fat, with the onset of the hypotensive response being earlier after glucose ingestion [17]. In addition, the type of protein may affect lowering of blood pressure, and certain proteins (e.g., in ancient wheat) have an effect on endothelial reactivity [25].

Protein supplements, usually in drink form, are increasingly administered to older people, particularly the institutionalized elderly, who are at a high risk of postprandial hypotension. While we have shown that protein loads of 70 g can lead to substantial falls in the SBP [18], this amount of protein is greater than usually taken at one time by older people. Lower doses are more likely to be used in supplements for older people, and there is evidence that ~30 g doses, once or twice daily, are likely to be beneficial for weight, nutrition, and muscle preservation [19]. Our observation suggests that care may need to be taken to monitor and prevent the hypotensive effects of protein supplements in susceptible older people, even when used in doses as low as 30 g and for 3 hours or even longer after protein supplement ingestion; our observations finished at 3 h when the SBP was possibly still reduced.

The heart rate (HR) increased after both whey protein drinks, with the greatest increase after the 70 g drink. These dose-responsive HR increases help to maintain cardiac output and hence the BP in the face of diversion of blood to the gut to aid protein absorption and digestion. The greater compensatory increase in the HR (and thus cardiac output) after the 70 g than the 30 g drink is likely to be a factor in the two doses lowering the BP to a similar degree. This might also suggest that if the compensatory HR increases after protein (or other nutrient) ingestion are impaired in older people, the decrease in the BP after nutrient ingestion will be greater and potentially more likely to lead to falls and other adverse effects. This is supported by our previous finding that in young men, there were greater increases in the HR and lesser decreases in the SBP than in older men after 70 g whey protein drinks [18]. Older people taking medications such as beta-blockers, which limit the heart rate and heart rate responses, or with cardiac conditions leading to bradycardia may be at even greater risk of post-protein BP falls. Following a standard mixed macronutrient breakfast meal, 16 subjects over 75 years old showed a significant fall in blood pressure, which may, in some less robust elderly persons, contribute to falls, while the younger control group did not show a significant decrease in postprandial blood pressure [26].

Our study has some limitations. The study was only conducted in healthy men, with a relatively small number of participants. The results may not be translatable to women. While our observation period of 3 hours was longer than in most previous studies, the BP was possibly still decreasing and the HR was still higher than on the control day 3 h after the protein drinks. A longer study duration would have helped determine the full time course of the effects of protein drinks. Since only whey protein was studied, the observation cannot be applied to other protein sources. A greater protein-induced BP fall than observed in this study might occur in older people at greater risk of postprandial hypotension than our subjects, for example, institutionalized, frail older people.

A possible safety issue for older people adopting a program of post-protein supplement or postprandial exercise might be excessive BP drops leading to falls. Our results suggest that if the intention is to give an older person a nutritional supplement drink containing 30–70 g of whey protein to preserve or even enhance muscle mass and function, excessive post-protein BP is a possibility, particularly in those most at risk. Consideration should be given to monitoring for this and/or advising measures (such as care when standing) to reduce the harmful effects of excessive postprandial BP decreases. It would also be appropriate in future studies to examine the effects on the BP in older people of combining protein and other nutrients with exercise.

## 5. Conclusions

The SBP decreased after both WP drinks but was not dose dependent compared to the control, with the effects most evident between 120 and 180 min. Accordingly, ingestion of even relatively modest protein loads in older men has the potential to cause PPH.

## Figures and Tables

**Figure 1 nutrients-14-01913-f001:**
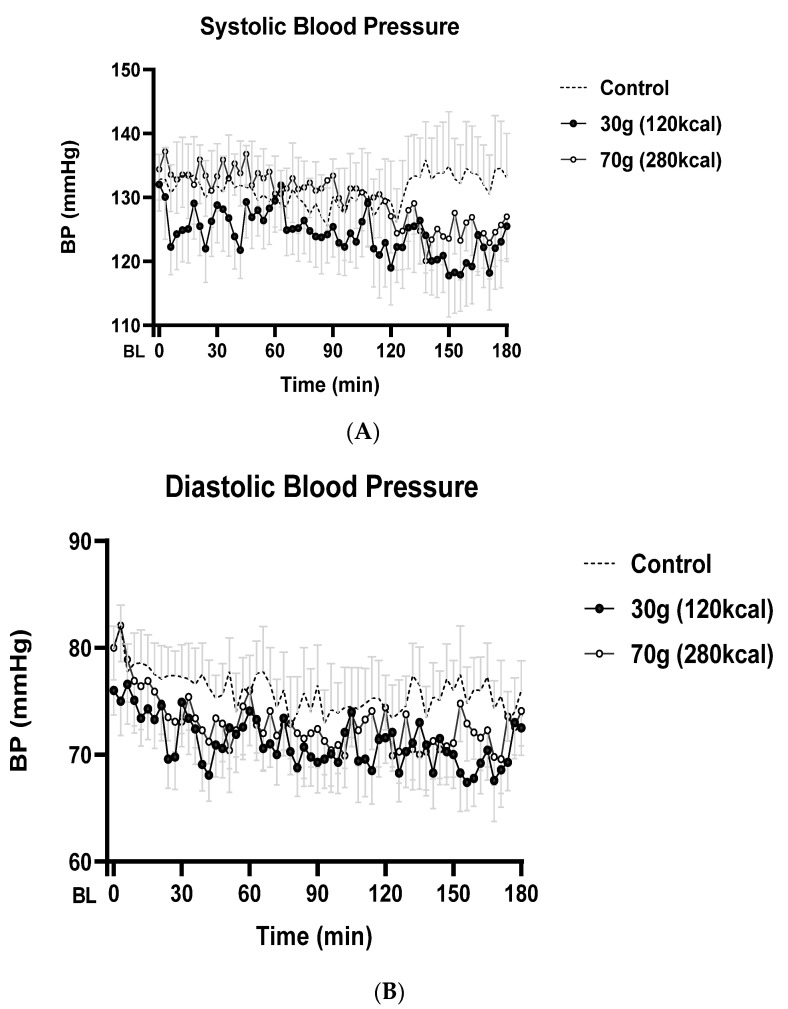
Mean (±SEM) systolic (**A**) and diastolic (**B**) blood pressure (mmHg) and heart rate (bpm) (**C**) following ingestion of drinks containing (i) flavored water (control, ~2 kcal) or (ii) 30 g (120 kcal) of whey protein (WP30) or (iii) 70 g (280 kcal) of whey protein (WP70) in older (*n* = 8) men. The baseline blood pressure represented as ‘BL’ in the figures was calculated as an average of measurements at −9, −6, and −3 min. T = 0 min refers to the point immediately after drink consumption. Drink condition effects were determined by using mixed model analysis. SBPs were lower after the whey protein drinks when compared to the control drink during the third hour (*p* < 0.05). The mean fall in the DBP was greater after 30 g and 70 g of whey protein drinks than after the control drink. The HR was higher after the whey protein drinks when compared to the control drink during the second (*p* = 0.007) and third (*p* = 0.005) hours, with the greatest increase after WP70 (*p* = 0.002).

**Table 1 nutrients-14-01913-t001:** Blood pressure and heart rate after whey protein drink ingestion.

	Control	WP30	WP70	*p*-Value
**SBP (mmHg)**				
Mean 0–60 min	132 ± 6	127 ± 4	134 ± 5	0.123
Mean 60–120 min	129 ± 6	124 ± 5	131 ± 6	0.092
Mean 120–180 min	133 ± 7	122 ± 6	125 ± 5	0.016
Max. change from baseline	−15 ± 2	−24 ± 2	−25 ± 4	0.020
Time to nadir (min)	90 ± 8	111 ± 22	136 ± 11	0.084
**DBP (mmHg)**				
Mean 0–60 min	77 ± 7	73 ± 2	75 ± 3	0.235
Mean 60–120 min	75 ± 3	71 ± 3	72 ± 3	0.414
Mean 120–180 min	75 ± 3	70 ± 23	72 ± 3	0.202
Max. change from baseline	−13 ± 2	15 ± 1	−17 ± 2	0.130
Time to nadir (min)	89 ± 19	98 ± 19	108 ± 11	0.740
**HR (bpm)**				
Mean 0–60 min	55 ± 3	58 ± 3	59 ± 3	0.057
Mean 60–120 min	53 ± 2	60 ± 3	60 ± 3	0.007
Mean 120–180 min	53 ± 2	58 ± 3	62 ± 3	0.005
Max. change from baseline Time to nadir (min)	2 ± 1 71 ± 25	6 ± 1 112 ± 17	10 ± 2 120 ± 17	0.002 0.181

Mean (±SEM) systolic blood pressure (SBP, mmHg), diastolic blood pressure (DBP, mmHg), and heart rate (beats per minute, bpm) following ingestion of drinks containing (i) flavored water (control, ~2 kcal) or (ii) 30 g (120 kcal) of whey protein (WP30) or (iii) 70 g (280 kcal) of whey protein (WP70) in older (*n* = 8) men. *p*-Value treatment effect of mixed model analysis.

## Data Availability

The data presented in this study are available on request from the corresponding author. The data are not publicly available due to ethical restrictions of the protocol having mentioned in our approved local ethical application that data will not be available for the general public.

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
