# Peer review of "Blood Pressure and Heart Rate Responses following Dietary Protein Intake in Older Men"

_nutrients, 2022, doi:10.3390/nu14091913_

Round 1

Reviewer 1 Report

Line 119: did the chairs have arms? (assuming this is a significant precaution given the nature of the study)

Line 25 you state maximum SBP drop occurs at the third hour but this does not seem to match the results section at line 155-160 stating 'maximum decreases occured at 1.5-2 hours- this might be my reading/interpretation but needs looking at/clarifying.

In previous work, I believe, you have looked at postprandial drops in BP follwing mixed macronutrient intake- why not comment (probably in discussion) on the drop resulting from 30/70g of whey in this study versus a mixed meal analysis in other work? i.e. give the reader an indication of whether protein alone seems to be worse/better than a mixture of carbs/protein/fat? 

You conclude that a protein supplement may pose a risk for falls but I really want to know (and I think your other readers will too) what is the risk following a regular/typical meal?

Reviewer 2 Report

Dear Editor,

I carefully read the manuscript by Oberoi et al.

My comments and suggestions for the authors are the following:

  • References should be formatted according to the instructions for the authors of the journal.
  • The authors should include a post-hoc power analysis. If the analysis is not powered, the authors should describe their study as pilot.
  • The results should be more appropriately reported in a table detailing between-groups differences in investigated parameters.
  • The authors should consider to refer to doi: 10.3390/nu10111666 in the manuscript.

Round 2

Reviewer 2 Report

Dear Editor,

I carefully read the revised version of the manuscript that is significantly improved in comparison with the original version. However, the authors should formatted the references according to the instructions of the Journal. The article presentation is not very accurate now.

This manuscript is a resubmission of an earlier submission. The following is a list of the peer review reports and author responses from that submission.

Round 1

Reviewer 1 Report

I congratuloate the authors on another interesting paper on a subject of great importance/significance to the health of older people. I have a few important and fairly minor suggestions: 

Line 24 (abstract and then throughout) you are describing change occuring in 50% of WP consumers versus 'only' 38% of controls etc- are percentages appropriate when we are talking about a study of 8 people? i.e. 4 people versus 3 people etc... it potentially inflates findings and would be better/clearer to just refer to numbers i.e. 3/8 etc.

Further studies

I can see ethical/safety concerns for people following a protocol where they ingest protein and then perform very light physical activity (which could be controlled perhaps by performing seated exercise)- but such activity certainly seems to effect post-prandial area under the curve for blood glucose- what about adding gentle physical movement to your studies on BP?

Conclusions

'SBP decreased after both WP drinks' suggest adding : 'but was not dose dependent'

Author Response

I congratulate the authors on another interesting paper on a subject of great importance/significance to the health of older people. I have a few important and fairly minor suggestions: 

Line 24 (abstract and then throughout) you are describing change occurring in 50% of WP consumers versus 'only' 38% of controls etc- are percentages appropriate when we are talking about a study of 8 people? i.e. 4 people versus 3 people etc... it potentially inflates findings and would be better/clearer to just refer to numbers i.e. 3/8 etc.

We thank the reviewer for his/her insightful comments and agree with the reviewer's point- have taken out the percentages and added numbers in lines 24-25 (abstract) and lines 159-160 (text)- SBP decreased ≥20mmHg in older men after both WP drinks (WP30: 6/8; WP70: 5/8) compared to 3/8 after control.

Further studies

I can see ethical/safety concerns for people following a protocol where they ingest protein and then perform very light physical activity (which could be controlled perhaps by performing seated exercise)- but such activity certainly seems to effect post-prandial area under the curve for blood glucose- what about adding gentle physical movement to your studies on BP?

The reviewer raises the important consideration that if older people exercise after protein ingestion there might be safety issues (eg excessive BP drops and increased risk of falls). We have now included mention of this in the discussion in lines 248-256 - A possible safety issue for older people adopting a program of post protein supplement or postprandial exercise might be excessive BP drops leading to falls. It would be appropriate in future studies to examine the effects on BP in older people of combining protein and other nutrients with exercise.   

Conclusions

'SBP decreased after both WP drinks' suggest adding: 'but was not dose dependent'

 We agree and have added the words 'but was not dose-dependent in line 259

Reviewer 2 Report

Within the article "Whey-protein load effects on blood pressure and heart rate in older men" the authors describe the physiological adaptions of blood pressure and heart rate on the ingestion of either 30 g whey protein, 70 g whey protein or a non-caloric control. The topic of the publication is interesting. However, the number of participants is small and the amount of results is very limited. Therefore, the paper seems more appropriate for a short communication or something comparable. Additionally, there is a high overlap with other publications of the authors where they have seemingly analysed the same persons together with other study participants. The choice of the analysed participants, parameters and the differences to the already published results should be depicted more clearly within the article. This was already mentioned at the end of the introduction but leaves a lot of questions (how was the subset chosen, why were the women excluded from the study.

The data and statistical analysis has to be written more clearly. The calculation of the baseline blood pressure and T0 should be included within the first paragraph. It is unclear if the values were checked for normality that would be a prerequisite for showing SEM and using ANOVA. It was stated that a repeated-measures ANOVA was used. However, the general time and treatment effect and the interaction between them was not reported. Therefore, it seems that every hour-period was compared separately.

Reviewer 3 Report

This study investigated the dose effects of whey-protein drink on blood pressure and heart rate in older men. The results showed that both 30 g and 70 g of whey-protein ingestion may decrease SBP and lead to postprandial hypotension in older men.

Comments:

This is a human study with a small sample size, short-period, and simple study design. The authors already published 4 papers regarding the effects of 30 and 70 g of whey-protein load on gastric emptying, appetite, energy intake, glucose homeostasis, and hormone in older and young people. In this manuscript, blood pressure and heart rate are the only two data. It seems like that this manuscript can be a short communication instead a full paper.

Author Response

This is a human study with a small sample size, short-period, and simple study design. The authors already published 4 papers regarding the effects of 30 and 70 g of whey-protein load on gastric emptying, appetite, energy intake, glucose homeostasis, and hormone in older and young people. In this manuscript, blood pressure and heart rate are the only two data. It seems like that this manuscript can be a short communication instead a full paper.

We thank the reviewer for his/her insightful comments. We are happy to proceed with this manuscript as a short communication.

Round 2

Reviewer 3 Report

This manuscript did not add any other data at all. Blood pressure and heart rate are the only two data. Still, it seems like a short communication instead a full paper.